# Hyaluronan and Derivatives: An In Vitro Multilevel Assessment of Their Potential in Viscosupplementation

**DOI:** 10.3390/polym13193208

**Published:** 2021-09-22

**Authors:** Annalisa La Gatta, Antonietta Stellavato, Valentina Vassallo, Celeste Di Meo, Giuseppe Toro, Giovanni Iolascon, Chiara Schiraldi

**Affiliations:** 1Department of Experimental Medicine, University of Campania “Luigi Vanvitelli”, Via De Crecchio 7, 80138 Naples, Italy; annalisa.lagatta@unicampania.it (A.L.G.); antonietta.stellavato@unicampania.it (A.S.); valentina.vassallo@unicampania.it (V.V.); celeste.dimeo@unicampania.it (C.D.M.); 2Department of Medical and Surgical Specialties and Dentistry, University of Campania “Luigi Vanvitelli”, 80138 Naples, Italy; giuseppe.toro@unicampania.it (G.T.); giovanni.iolascon@unicampania.it (G.I.)

**Keywords:** intra-articular injection, hyaluronan, rheology, hyaluronidase, human sinoviocytes, human chondrocytes, OA biomarkers

## Abstract

In this research work, viscosupplements based on linear, derivatized, crosslinked and complexed HA forms were extensively examined, providing data on the hydrodynamic parameters for the water-soluble-HA-fraction, rheology, sensitivity to enzymatic hydrolysis and capacity to modulate specific biomarkers’ expression in human pathological chondrocytes and synoviocytes. Soluble HA ranged from 0 to 32 mg/mL and from 150 to 1330 kDa MW. The rheological behavior spanned from purely elastic to viscoelastic, suggesting the diversity of the categories that are suitable for restoring specific/different features of the healthy synovial fluid. The rheological parameters were reduced in a diverse manner upon dilution and hyaluronidases action, indicating different durations of the viscosupplementation effect. Bioactivity was found for all the samples, increasing the expression of different matrix markers (e.g., hyaluronan-synthase); however, the hybrid cooperative complexes performed better in most of the experiments. Hybrid cooperative complexes improved COLII mRNA expression (~12-fold increase vs. CTR), proved the most effective at preserving cell phenotype. In addition, in these models, the HA samples reduced inflammation. IL-6 was down-regulated vs. CTR by linear and chemically modified HA, and especially by hybrid complexes. The results represent the first comprehensive panel of data directly comparing the diverse HA forms for intra-articular injections and provide valuable information for tailoring products’ clinical use as well as for designing new, highly performing HA-formulations that can address specific needs.

## 1. Introduction

Osteoarthritis (OA) is a progressive degenerative disorder characterized by a breakdown of the cartilage in the joints, a deterioration of the synovial fluid and a subchondral osteosclerosis accompanied by osteophyte formation [1]. The prevalence of OA increases with age, with the majority of people over 60 years of age presenting certain cartilage abnormalities. Current studies show that synovitis, low-grade inflammation, and remodeling of the subchondral bone can precede cartilage deterioration [2,3].

Specifically, the synovial fluid composition is modified due to degradation of hyaluronan and/or dilution by inflammatory effusion [4,5]. As a consequence, viscoelasticity drops expose joints to damage/wear, causing pain and reductions in joint mobility [6]. Viscosupplementation, involving (repeated) intrarticular injection of hyaluronan-based formulations, currently represents the most employed, not invasive, symptomatic treatment of the disease [7]. The delivery of HA-based formulations into the joint is intended to restore the rheological properties of the synovial fluid, resulting in pain relief and improved joint mobility [8,9,10,11,12,13,14]. Furthermore, the biochemical effect of HA on chondrocytes and synoviocytes may permit at least a partial restoration of physiological function, towards a certain grade of repair [15] A huge number of HA-based products intended for intrarticular injections are available on the market. However, the scientific evidence regarding these products is still quite limited [16]. The information included in the leaflets is incomplete and, in most cases, not supported by either in vitro or in vivo studies, and relevant differences among them are rarely reported [16]. Rheological properties are mainly analyzed to evaluate the viscosupplementation potential, and reports comparing the performance of diverse formulations are available [10,17,18]. Furthermore, in vitro or animal studies describe results related to the biological features of the preparations without referring those to the rheological properties of the gels/products [18,19,20,21]. Clinical trials are available for few of the preparations [16,22,23,24,25,26,27,28,29]. However, these studies very rarely, or almost never, compare different medical devices. Availability of in vitro and or in vivo data comparing these injective medical devices in the same experimental set-up is strongly needed to scientifically support clinicians’ awareness in the selection of the product most suitable for each specific clinical case (different stages/patterns of OA) [16]. The increasing number of commercialized products (EMA- or FDA-approved and/or registered in Asia and Africa), does not permit a full array of comparisons between all of them. Comparisons among categories of products may be considered. One of the main aspects on which these preparations differ is the HA chemical form. In fact, depending on the kind of hyaluronan used, three categories of formulations may be identified, namely: (1) linear HA; (2) chemically modified HA (derivatized or crosslinked); and (3) HMW/LMW HA hybrid complexes. The first category comprises HA solutions in physiological medium that differ in terms of HA concentration, molecular weight and source [11,17,18,30,31,32,33]. The second category mainly consists of or chemical hydrogels (not solutions) that may also contain a certain amount of water-soluble HA chains [34,35,36,37,38,39,40]. The third group includes the recently developed physical complexes based on hydrogen cooperative bonds between HA chains of diverse sizes [15,18,29,41,42] We aimed to provide scientific evidence of the diverse biophysical and biochemical properties of these three main categories of products by carrying out an in vitro characterization of preparations belonging to each of the groups indicated above. 

Characterization comprised hydrodynamic and rheological analyses, which aimed to quantify and characterize the water-soluble HA included in the formulas and the viscosupplentation potential along with its variation due to dilution and HA-enzymatic hydrolysis (both phenomena occur in vivo after delivery). Moreover, we compared their bioactivity in a human primary cell model of OA, analyzing the cell morphology, viability and biosynthesis of key biomarkers of cartilage, and also evaluating specific mediators to address the inflammatory status [43,44]. To the best of our knowledge, this is the first study directly comparing both the biophysical and biochemical features of products for intrarticular injections based on the diverse chemical forms of hyaluronan.

## 2. Materials and Methods

### 2.1. Materials

Hyalubrix^®^ and HyMovis^®^ HYADD4 (Fidia Farmaceutici S.p.A., Abano Terme, Padova, Italy), Sinovial HL^®^ (IBSA Farmaceutici Italia S.r.l, Lodi, Italia) and Jonexa Hyalastan SGL-80™ Hyalastan SGL-80™ (Genzyme Corporation, Pleasant View Terrace, Ridgefield, NJ, USA) were commercially available in the local pharmacy. Information on the gels, as reported in the package inserts, is shown in Table 1. Hyalubrix^®^ is a sterile non-pyrogenic, fermentative linear HA solution. HyMovis^®^ is an 8 mg/mL formulation in PBS of HYADD4^®^, an HA amide derivative, where 2–3% of the carboxyl groups of the polymer are modified by introducing hexadecylamine as side chains via the amide bond [12]. This HA amide derivative is a partially hydrophobized HA. It is reported to form a physical hydrogel in aqueous medium [38]. Jonexa Hyalastan SGL-80™ is reported, by the manufacturer, to be a mixture of divinyl sulfone (DVS)-crosslinked HA gel and sodium hyaluronate fluid. Sinovial HL^®^ is a HMW/LMW HA-based preparation. Bovine testicular hyaluronidase, BTH (EC 3.2.1.35), salt-free lyophilized powder with a specific activity of 890 U/mg, was purchased from Sigma-Aldrich S.r.l. (Milan, Italy) (cat. N. H3884, lot. SLBL6343V). Dulbecco’s Phosphate Buffered Saline (PBS) without calcium and magnesium was purchased from Corning, USA (ref. 21-031-CV). Human primary antibodies against COMP-2 and HAS-1, used for immunofluorescence staining, were purchased from Abcam, Cambridge, MA, USA, and Santa Cruz Biotechnology, Dallas, TX, USA, respectively. FITC-conjugated goat anti-rabbit secondary antibody was purchased from Life Technologies, Milano, Italy. Fluorescent phalloidin TRITC conjugate was purchased from Sigma-Aldrich, Italy, and Mounting Medium with a DAPI-aqueous Fluoroshield was purchased from Prodotti Gianni S.r.l., Milano, Italy. Dulbecco’s Modified Eagle’s Medium (DMEM) and all cell’s reagents were obtained from Gibco (Thermofisher Scientific, Waltham, MA, USA). Cell viability reagent was CCK-8 and was purchased from Dojindo EU GmbH. TRIzol^®^ Reagent was obtained from Invitrogen, Milan, Italy. Paraformaldehyde and triton X-100 were purchased from Sigma Aldrich Milan, Italy. Oligonucleotides primer sequences were synthesized from TibMolBiol S.r.l, Genoa, Italy.

### 2.2. Methods

#### 2.2.1. Water-Soluble HA Quantification and Hydrodynamic Analyses

Water-soluble HA quantification and its hydrodynamic analysis were accomplished using the size exclusion chromatography-triple detector array (SEC-TDA), as previously reported, with slight modification [45,46]. Briefly, samples were diluted in water to a 4 mg/mL final concentration and kept overnight under stirring (900 rpm, 37 °C). The samples were then centrifuged (13,000× *g*, 5 min) and the supernatant were filtered on 0.22 µm disposable membranes. Filtered samples (containing the water-soluble HA fraction) were opportunely diluted for Size Exclusion Chromatography-Triple Detector Array (SEC-TDA) analyses using the Viscotek system (Viscotek, Malvern, UK). A detailed description of the SEC-TDA system, of its potential to analyze biopolymers such as HA, and of the analysis conditions, are reported elsewhere [45,46]. Biopolymer (water-soluble HA fraction) concentration (mg/mL), and such hydrodynamic parameters as molecular weight (M_w_, M_n_, M_w_/M_n_), molecular size (hydrodynamic radius—R_h_), and intrinsic viscosity (η) distributions, were derived.

#### 2.2.2. Rheological Characterization

Rheological evaluation of the gels was carried out using a Physica MCR301 oscillatory rheometer (Anton Paar, Ostfildern-Scharnhausen, Germany). Oscillation and steady shear measurements were performed using a CP 50-2 geometry (cone diameter 49.970 mm, cone angle 1.995°, truncation 207 µm). All measurements were performed at 37 °C. Amplitude sweep tests were carried out at 1.59 Hz frequency (that is 10 rad s^−1^) over a strain amplitude range of 0.01–100%, with 50 measuring points and no time setting. The linear viscoelastic range (LVR) was derived as the range of amplitude in which the moduli remained constant. Oscillation frequency sweep tests were then carried out over a frequency range of 0.159–15.9 Hz (that is 1–100 rad·s^−1^), at a constant strain selected within the LVR (2%), 10 measuring points/dec. The values for the moduli and complex viscosity at 0.5 and 2.5 Hz (physiological walking and running frequency) were derived. Measurements were run in duplicate. Gels were characterized also after 1:2 (*v*/*v*) dilution in physiological medium. Flow curves were obtained by recording the dynamic viscosity of the samples as a function of shear rate (0.001–1000 s^−1^), 50 measuring points, and no time setting.

#### 2.2.3. Sample Rheology during Enzymatic Degradation

The rheological parameters (G′, G″, and complex viscosity) for the gels were observed and measured during incubation with BTH. Specifically, a BTH solution in PBS was added to each formulation to obtain a final BTH concentration of 10 U/mL and a gel dilution ratio of 1:2 (*v*/*v*). Samples were gently mixed and rapidly placed on the lower plate of the rheometer. Storage, loss moduli, and complex viscosity were measured as a function of time while the strain amplitude and the frequency were kept constant at 2% and 2.5 Hz, respectively. Measurements were performed at 37 °C, and the gel and the BTH solutions were equilibrated at that temperature before starting the experiment. The same test was performed for each gel after the same dilution with PBS (control). Attention was given to ensure similar testing conditions for all the samples. Rheological measurements started after approximately 5–6 min from the addition of the enzyme. The decrease in dynamic parameters due to enzymatic hydrolysis was monitored, and the residual value for complex viscosity at specific incubation time intervals was calculated (% compared to control) up to 30 min incubation as follows:Residual viscosity (%) = (viscosity (t))/(viscosity (t = 0)) × 100(1)
where viscosity (t) is the viscosity value at 2.5 Hz at a certain incubation time and viscosity (t = 0) is the viscosity value at 2.5 Hz measured for the control. Analyses were performed in duplicate. Results are reported as the mean value ± SD.

#### 2.2.4. In Vitro Model of Osteoarthritis for Comparative Evaluation of Diverse HA-Based Injective Hydrogels

Primary in vitro cultures of pathological human chondrocytes and synoviocytes were obtained following the experimental protocols that are well established in our laboratories [15,44]. In detail, the knee cartilages specimens and synovium samples were obtained at the Department of Medical and Surgical Specialties and Dentistry, University of Campania “Luigi Vanvitelli”, Naples, (Italy), from OA-affected patients who were subjected to knee surgery. The isolation and use of primary human cells was formally approved by the Internal Ethical Committee (AOU-SUN registration no. 0003711/2015). Cartilage tissue was digested through a type I collagenase at 3 mg/mL and a dispase-based solution at 4 mg/mL, at a temperature of 37 °C, on a shaking plate for 16 h. Then, the cell suspension was filtered (70 μm, BD, Falcon Glendale, AZ, USA) and centrifuged at 1500 rpm for 7 min. Cellular pellet was resuspended in DMEM supplemented with Fetal Bovine Serum (FBS) (10% *v*/*v*), penicillin-streptomycin (1% *v*/*v*), and Amphotericin B (1% *v*/*v*). The cells were seeded in a 24-well plate and maintained at 37 °C in a humidified atmosphere with 5% *v*/*v* CO_2_, changing the culture medium every 48 h. The synovial fluid was centrifuged and the pellets that contained the synoviocytes were reseeded. After 2 weeks of in vitro culture, the chondrocytes and synoviocytes were harvested with trypsin/EDTA 0.2 mg/mL and reseeded into a 24-well plate, 2.0 × 10^4^ cells/cm^2^ to start with the comparative treatments. Phenotypic characterization of articular chondrocytes and synoviocytes were performed through Fluorescence-Activated Cell Sorting (FACS) as previously reported [15].

##### Human Primary Cell Cultures Setting Up

Primary in vitro cultures of pathological human chondrocytes and synoviocytes were obtained following the experimental protocols that are well established in our laboratories [15,44]. In detail, the knee cartilages specimens and synovium samples were obtained from the Department of Medical and Surgical Specialties and Dentistry, University of Campania “Luigi Vanvitelli” (Naples, Italy), from OA-affected patients who were subjected to knee surgery. The isolation and use of primary human cells was formally approved by the Internal Ethical Committee (AOU-SUN reg. no. 0003711/2015). Cartilage tissue was digested through a collagenase type I at 3 mg/mL and dispase-based solution at 4 mg/mL, at 37 °C temperature, on a shaking plate for 16 hr. Then, the cell suspension was filtered (70 μm, BD, Falcon, USA) and centrifuged at 1500 rpm for 7 min (Eppendorf Centrifuge). The cellular pellet was resuspended in DMEM, supplemented with Fetal Bovine Serum (FBS) (10% *v*/*v*), penicillin-streptomycin (1% *v*/*v*), and Amphotericin B (1% *v*/*v*). The cells were seeded in a 24-well plate and maintained at 37 °C in a humidified atmosphere with 5% *v*/*v* CO_2_, changing the culture medium every 48 h. The synovial fluid was centrifuged and the pellets that contained the synoviocytes were reseeded. After 2 weeks of in vitro culture, the chondrocytes and synoviocytes were harvested with trypsin/EDTA 0.2 mg/mL and split again into a 24-well plate, 2.0 × 10^4^ cells/cm^2^, to begin the comparative treatments. Phenotypic characterizations of articular chondrocytes and synoviocytes were performed through Fluorescence-Activated Cell Sorting (FACS), as previously reported [15].

##### Viability Assay

Primary cells, isolated and seeded as described above, were incubated in the FBS-supplemented DMEM (untreated pathological cells, CTR), or in the presence of the following HA-based gels: Hyalubrix^®^, HyMovis^®^, Jonexa Hyalastan SGL-80™, and Sinovial HL^®^. These products were used at a final dilution of 1:4 in the culture medium. The cells were treated for 24 h; afterwards, their viability was assessed using the Cell Counting Kit-8 (Dojindo EU GmbH, München, Germany) following the manufacturer’s protocol. The optical densities of the obtained solutions were measured at 450 nm using a Beckman DU 640 spectrometer (Beckman, Milan, Italy). The relative cell viability was calculated as a percentage of the maximal absorbance:Viability = (100 × mean OD treated cells)/(mean OD control)(2)

##### Gene Expression Analyses of OA Biomarkers through qRT-PCR

After a 24-h incubation, the cells were harvested and lysated using TRIzol^®^ Reagent (Invitrogen, Milan, Italy) in order to isolate cellular RNA [47]. For each sample, 1 µg of total RNA was reverse transcribed into cDNA following the manufacturer’s protocol (Reverse Transcription System Kit, Promega, Milan, Italy). In this way, a quantitative Real-Time PCR was performed by the IQ™ SYBR^®^ Green Supermix (Bio-Rad Laboratories, Milan, Italy). The specific primer sequences used for these analyses are reported in Table 2. All the samples were analyzed in triplicate, and the glyceraldehyde-3-phosphate dehydrogenase (GAPDH) housekeeping gene was used in order to normalize the mRNA expression of analyzed genes. The variations were calculated using the comparative threshold method (∆∆Ct = difference in ∆Ct between GAG-treated cells and control), and the results are reported as the normalized fold expression using the quantification of 2^−ΔΔCt^ method [48] and Bio-Rad iQ5 software (Bio-Rad Laboratories, Milan, Italy).

##### Western Blot Analyses Performed on Pathological Chondrocytes

For specific protein expression analyses, chondrocytes were treated with HA-based gels for 48 h. In detail, after the treatments, each sample was lysed using a Radio-Immunoprecipitation Assay (RIPA buffer 1×; Cell Signaling Technology, Danvers, MA, USA) and intracellular protein concentrations were evaluated by protein assay reagent. In this way, 120 ng of proteins were loaded by SDS-PAGE 10% polyacrylamide gel. Then, the proteins were transferred to nitrocellulose membrane (GE, Amersham, White Lion Rd, Little Chalfont, UK) for 1 h at 110 V and the latter was then blocked with 5% non-fat milk in Tris-buffered saline and 0.05% Tween-20 (TTBS) for 30 min. The membrane was incubated with primary antibodies against COMP-2 (Santa Cruz Biotechnology, Dallas, TX, USA, used at a dilution ratio of 1:500) and NF-kB (Santa Cruz Biotechnology, Dallas, TX, USA, used at a dilution ratio of 1:250) overnight at 4 °C. On the following day, the membrane was washed three times for 10 min with TTBS and incubated with horseradish peroxidase-conjugated anti-mouse and anti-rabbit antibodies (Santa Cruz Biotechnology, Dallas, TX, USA, diluted at 1:5000). Blots were developed through the ECL system (Merck KGaA, Darmstadt, Germany,) and tubulin antibody (Santa Cruz Biotechnology, Dallas, TX, USA, diluted at 1:1000) was used as a gel loading control. Finally, the semi-quantitative protein expression analysis was performed using Image J software.

##### Immunofluorescence Analyses

After 24 h, supernatants were removed from the cell monolayer. Then, the treated and untreated primary cells were washed twice with PBS and fixed with paraformaldehyde 4% *w*/*v*. Following that, a solution of Triton X-100 at 0.2% *v*/*v* in PBS was used as a cellular permeabilizer. Immunofluorescence was performed following the experimental protocol previously reported [47]. In this context, human primary antibodies against COMP-2 (diluted 1:100), and HAS-1 (diluted 1:100) were incubated overnight at 4 °C. Then, an FITC-conjugated goat anti-rabbit secondary antibody was used at a dilution ratio of 1:1000. Fluorescent phalloidin TRITC conjugate (50 µg/mL) was used to stain actin filaments. Slides were covered using Mounting Medium with DAPI-aqueous Fluoroshield, and specific images were obtained using a fluorescence microscope Axiovert 200 (Zeiss, Milan, Italy) and analyzed using AxioVision 4.8.2. A second experiment was performed by specifically staining the proteins of interest (COMP-2 or HAS-1) and the nuclei.

## 3. Results

### 3.1. SEC-TDA Analyses

The results of the SEC-TDA analyses are reported in Table 3. The water-soluble HA concentration values measured for the first and third group products were in agreement with the values reported in the package inserts (Table 1) [33,38,40,41]. Analyses performed for the chemically modified HA samples revealed that Jonexa Hyalastan SGL-80™ presented a water-soluble HA fraction corresponding to less than 2% of total HA. No water-soluble polymeric chains could be detected for HyMovis^®^ in our experimental conditions. The hydrodynamic parameters indicated the presence of high molecular weight HA in the unmodified HA-based samples (first and third group) with a polydispersity index in the range of 1.2–1.4. The soluble HA fraction in Jonexa Hyalastan SGL-80™ consists of a slightly wider distribution of high molecular weight HA chains as well (M_w_ around 1300 kDa; M_w_/M_n_ = 1.6). Intrinsic viscosity and hydrodynamic radius values are consistent with values reported in the literature for comparable molecular weight samples [45]. The size exclusion chromatography applied could only partially resolve the hybrid complexes. In fact, SEC-TDA analyses for Sinovial HL^®^ are hampered by the cooperative hydrogen bonds between the two entangled fractions that are stabilized through thermal treatment. We thus report, in the table, the so-called “apparent MW” values, which are, in this case, analyzed taking into account the laser and viscosity signals.

### 3.2. Rheological Characterization

Figure 1 shows the mechanical spectra for the preparations as distributed in the single dose syringes and after 1:2 dilution in the physiological medium. Hyalubrix (1st group) and Sinovial HL^®^ hybrid complexes (3rd group) samples showed a typical viscoelastic behavior with G″ exceeding G′ at low frequencies and with G′ increasing with frequency more markedly than G″, until a crossover was observed. At higher frequency values, an elastic behavior was recorded. The cross-over frequency for the samples was in the range of 3–7 Hz. Chemically modified HA samples (2nd group) behaved differently. For Jonexa Hyalastan SGL-80™, a slight increase in the dynamic moduli was observed with the increasing of the frequency. HyMovis^®^ exhibited diverse moduli variation. G′ increased, albeit less markedly than for Jonexa Hyalastan SGL-80™, while G″ was not dependent on frequency. The dilution had a different effect on the diverse groups of formulations. Moduli decreased by approximately 10- to 20-fold at 2.5 and 0.5 Hz frequency for the first and third group samples, and a shift of the crossover frequencies to higher values (around 10–12 Hz) was recorded. For the chemically modified samples, the moduli variation with frequency remained the same after dilution, while Storage Modulus decreased by around 4-fold and 2-fold for HyMovis^®^ and Jonexa Hyalastan SGL-80™, respectively, at both 0.5 and 2.5 Hz. The curves of tan δ and complex viscosity (η *) as a function of frequency are shown in Figure 2. For the first and third group samples, consistently with the mechanical spectra, a continuing and comparable reduction in tan δ was recorded while the frequency trend remained consistent with the undiluted samples. Among the modified HA preparations, tan δ varied diversely with frequency. HyMovis^®^ showed a reduction in tan δ with the increasing of the frequency while, for Jonexa Hyalastan SGL-80™, elasticity remained nearly the same regardless of the frequency. Differently from the other formulations, for the second group samples, tan δ was not affected by dilution. Variation of complex viscosity with frequency is reported in Figure 2a’–c’). Sinovial HL^®^ and Hyalubrix^®^ showed viscosity values around of 10 Pa × s at low frequencies with higher values for Hyalubrix^®^. Following dilution, approximately 11- and 8 to 9-fold decreases in viscosity were recorded at 0.5 and 2.5 Hz frequencies, respectively. The decrease was less marked when measured in the high-frequency region. The trend suggested a pseudoplastic behavior for these samples. This trend was confirmed by the steady shear measurements with zero-shear viscosity values resembling the ones in the oscillatory regime, and a shear thinning behavior starting from a shear rate of approximately 1 s^−1^. Both the chemically modified preparations of the second group showed a decrease in viscosity with frequency and were almost equivalent when compared in their commercialized form. However, the viscosity of the samples was diversely affected by dilution. In particular, a far higher viscosity reduction was recorded for HyMovis^®^, which exhibited an approximately 15- and 6-fold lower viscosity at 0.5 and 2.5 Hz frequency, respectively. Moreover, for HyMovis^®^, the variation of viscosity with the frequency after dilution resembled one of the linear and hybrid samples. Consistently, in steady shear measurements, viscosity constantly decreased with the shear rate and values in the range of 500–1000 Pa × s were measured at the lower shear rates tested. These values were much higher than those for the first and third group.

### 3.3. Sample Sensitivity to Enzymatic Degradation: A Rheology-Based Comparison

Rheological parameters were monitored to compare gel stability during incubation with BTH. Figure 3 shows the complex viscosity values recorded for the three group samples under enzymatic hydrolysis conditions compared to the control (samples incubated with PBS in place of the BTH solution). All samples showed complex viscosity remaining constant over time when diluted with PBS (during observation). When incubated with the BTH solution, the first and third group samples decreased their viscosity over time, indicating sensitivity to enzymatic hydrolysis. However, the rate of viscosity reduction was different with Hyalubrix^®^ degrading faster and Sinovial HL^®^ preserving its viscosity to a higher extent. After 10 min of incubation, Hyalubrix^®^ retained 27 ± 3% of its viscosity, while for Sinovial HL^®^, the residual viscosity was around 70%. Prolonging incubation to 25 min, Sinovial HL^®^ still maintained around 33% viscosity, while for Hyalubrix^®^, the viscosity was less than 10% compared to the control (dilution in the absence of BTH). Among the chemically modified HA-formulations, Jonexa Hyalastan SGL-80™ showed sensitivity to degradation under the applied conditions, while a peculiar result was obtained for HyMovis^®^. In this latter case, no variation in viscosity was observed after 5 min of incubation (around 100% viscosity, compared to control). Then, viscosity increased up to 150% at 10 min of incubation. No further variation in sample viscosity was then recorded in the frame time of the experiment. Differently from the other samples, Jonexa Hyalastan SGL-80™ exhibited a diverse degradation profile; the viscosity was reduced to a great extent during the first 5 min of incubation (60% residual viscosity), while further reductions occurred at a slower rate (46 ± 7% residual viscosity at 25 min of incubation). Compared to the cross-linked sample, Hyalubrix^®^ and Sinovial HL^®^ viscosity varied more gradually, and Sinovial HL^®^ showed a similar viscosity reduction to a cross-linked sample for up to 20 min of incubation.

### 3.4. In Vitro Model of Osteoarthritis for Comparative Evaluation of Diverse HA-Based Injective Hydrogels

#### 3.4.1. Cell Viability Assay in Presence of HA-Based Gels

As reported in Figure 4, none of the tested gels were cytotoxic for the cells. Specifically, in the chondrocytes, all the HA-based formulations sustained cell viability without significant differences. However, for the synoviocytes cultures, the addition of Jonexa Hyalastan SGL-80™ (1:4 dilution) reduced viability in comparison to the other samples. The result was statistically significant in comparison to Sinovial HL^®^-treated cells (*p* < 0.05). Moreover, in the pictures panel (Figure 4a,c), it is possible to note that cellular morphology was consistent with sound physiological conditions. On the other hand, HyMovis^®^-treated chondrocytes and synoviocytes were not visible since, due to product insolubility, even the gel diluted in the culture medium was not transparent and, thus, did not allow optical microscopy images of the cells. Nevertheless, viability tests proved that it did not hamper the growth of either chondrocytes and synoviocytes.

#### 3.4.2. Gene Expression Analyses of OA Biomarkers by qRT-PCR

COLII is the specific biomarker of the chondrocyte phenotype. In our results, COLII mRNA is significantly up-regulated with respect to CTR (untreated pathological cells) for the HA-based hydrogels tested. In particular, Sinovial HL^®^ was the most effective with respect to CTR and to all the others in terms of increasing COLII expression (approximately a 12-fold increase, * *p* < 0.01). Notably, HyMovis^®^ also increased COLII expression (approximately 10-fold). In addition, we evaluated HAS-1, which is one of the enzymes responsible for hyaluronan synthesis in cartilage tissue. As shown in Figure 5a, all samples increased HAS-1 expression with respect to CTR. However, the highest modulation was found for Sinovial HL^®^, with an approximately 14-fold increase, while the lowest was obtained by Hyalubrix^®^ treatment, whose increment was 2-fold compared to CTR. MMP-13 was also analyzed as a degradative matrix enzyme that is generally up-regulated in degenerative OA disease [21]. Interestingly, all of the tested samples down-regulated MMP-13 expression in comparison to CTR (all reported values were below 1). AGN and SOX-9 were also quantified in order to evaluate additional biomarkers with a specific role in osteoarthritis pathways. In particular, AGN mRNA was increased by Sinovial HL^®^ and Hyalubrix^®^, both approximately 1.5-fold vs. CTR, but its expression decreased in the presence of HyMovis^®^. Instead, for Jonexa Hyalastan SGL-80™ treatments, the AGN expression was similar to CTR. In addition, SOX-9, which is the main transcription factor of chondrocytes, was up-regulated, by Sinovial HL^®^, Jonexa Hyalastan SGL-80™, and HyMovis^®^, approximately 3-, 2-, and 1.5-fold, respectively. In the presence of Hyalubrix^®^, SOX-9 was down-regulated (Figure 5c). In addition, to evaluate the anti-inflammatory effects of hyaluronan-based gels, different pro-inflammatory biomarkers, such as COMP-2, IL-6, and TNF-α, were assayed. As shown in Figure 5e, all cytokines were down-regulated compared to CTR, proving the anti-inflammatory efficacy of HA samples. Overall, Sinovial HL^®^ was the most effective in terms of lowering cytokine expression.

In the synoviocyte cultures, HAS-1 was up-regulated, by Sinovial HL^®^, approximately 3-fold with respect to CTR (untreated pathological cells), as shown in Figure 5b. On the contrary, Jonexa Hyalastan SGL-80™, HyMovis^®^, and Hyalubrix^®^ reduced the expression of HAS-1 with respect to CTR. As was also the case for chondrocytes, MMP-13 proved to be down-regulated for synoviocytes by all of the tested HA gels (all reported values are below 1). Additionally, AGN and COL-I were also analyzed, as shown in Figure 5d. The gene expression results showed that AGN is up-regulated by Sinovial HL^®^ and HyMovis^®^ approximately 10-fold with respect to CTR. In the presence of Jonexa Hyalastan SGL-80™, its expression increased 12-fold, while with Hyalubrix^®^ treatment, AGN was slightly down-regulated (it was similar to CTR). In addition, in this cellular model, COL-I was up-regulated by Jonexa Hyalastan SGL-80™ 14-fold compared to the CTR level. It was slightly up-regulated with HyMovis^®^ and Hyalubrix^®^ HA-based gels. Notably, COL-I expression was down-regulated in the presence of Sinovial HL^®^. Furthermore, in the synoviocytes model, the anti-inflammatory effects of HA samples were tested. In particular, COMP-2, IL-6, and TNF-α were assayed. As shown in Figure 5f, all biomarkers were down-regulated with respect to CTR, proving the anti-inflammatory efficacy of these commercial HA-based gels, notwithstanding the presence of chemically modified structures. Only Hyalubrix^®^ reported a slight, but not significant, up-regulation (1.1-fold vs. CTR) of IL-6. Moreover, in the synoviocytes model, in all cases, Sinovial HL^®^ was shown to modulate the cytokine expression and COMP-2 towards a more physiological condition

#### 3.4.3. Western Blot Analyses Performed on Pathological Chondrocytes

To evaluate the effects of HA-based hydrogels, OA-related expression proteins were evaluated on pathological chondrocytes. In this context, COMP-2 and NF-kB were considered as biomarkers for analysis. Untreated cells (CTR) presented a marked expression of both of the biomarkers considered, thus confirming an ongoing process of inflammation and cartilage damage. The results showed that only Jonexa Hyalastan SGL-80™ and Sinovial HL^®^ reduced COMP-2 protein expression 1.21-fold and 1.98-fold (*p* < 0.05), respectively, in comparison to CTR (Figure 6). Moreover, NF-kB was down-regulated by all HA-based treatments, and Sinovial HL^®^, Jonexa Hyalastan SGL-80™ and also Hyalubrix^®^ proved more effective with respect to HyMovis^®^. NF-kB protein expression was reduced approximately 1.72-fold (*p* < 0.05), 1.20-fold and 1.21-fold with respect to CTR, in the treatments with Sinovial HL^®^, Jonexa Hyalastan SGL-80™ and Hyalubrix^®^ (Figure 6).

#### 3.4.4. Immunofluorescence Analyses

In the chondrocytes cell model, COMP-2 protein expression was investigated using immunofluorescence as a further marker of the ongoing pathological and inflammatory condition (OA in vitro model). As shown in Figure 7a, untreated pathological cells expressed a basal level of this biomarker proving an ongoing inflammation state and damage occurrence. The IF images recorded for treated cells proved that all these hydrogels were effective in COMP-2 reduction. A very low extent of green fluorescence was found in Sinovial HL^®^-treated samples, in agreement with the qRT-PCR analyses of other biomarkers. On the contrary, Jonexa Hyalastan SGL-80™- and HyMovis^®^-treated cells showed a more intense green signal evidencing lower efficacy to counteract the inflammatory condition (as shown in Figure 7a). In parallel, HAS-1 fluorescence intensity was analysed to evaluate the potential improvement of HA neo-synthesis in synoviocytes. Figure 7), proved that untreated cells did not express this biomarker, on the contrary, Sinovial HL^®^ treatment seemed to slightly increase its production in comparison to the other ones. In addition, Jonexa Hyalastan SGL-80™-, HyMovis^®^- and Hyalubrix^®^-treated cells showed reduced proliferation, as highlighted by the cell densities in the relative images. The Appendix A show the results of the immunofluorescence performed in both cellular models, with specific staining of the proteins of interest (COMP-2 or HAS-1 both in green) and nuclei. Again, Sinovial HL^®^ proved the most effective in terms of reducing COMP-2 and increasing HAS-1 (Appendix A, respectively).

## 4. Discussion

Several literature reports concern the rheological comparison of HA-based formulations intended for intra-articular use. Some in vitro or animal studies better describe the biological features of the preparations rather than the rheological properties. As a matter of fact, many different meta-analyses agree with regard to the beneficial effect of intra-articular injection of HA. However, the clinical studies were run in diverse countries, with perhaps similar, but not identical, products or protocols. Very rarely or almost never, different medical devices are compared in clinical studies. More frequently, they are injected parallel to the placebo, or pharmacological treatments are compared to HA-injective treatments [49]. In this respect, a better understanding of the specific features of each particular HA-based product is of key importance in order for the clinician to select the best performing one, which is suitable or ideal for treatment of a specific type of joint damage. In fact, different pathologies (e.g., age, obesity, sport injury) may require different treatments [50]. In this research work, we evaluate, by setting up a multi-level approach, widely used HA-based intra-articular formulations that were commercialized in Italy (Europe) and belong to three different categories, namely, linear HA (first group)-, cross-linked or derivatized HA (second group)-, and, finally, hybrid cooperative HA complex (third group)-based formulas.

Rheological analyses were performed to evaluate the products’ relative viscosupplementation potential. Data evidenced that the three group formulas covered rheological behavior spanning from viscoelastic to purely elastic, reflecting the diverse biopolymer structure, molecular weight, and concentration. In fact, as expected for crosslinked polymeric networks, Jonexa Hyalastan SGL-80^TM^ behaves as a pure gel with tan delta’s frequency remaining constant, as shown in Figure 2b [5,12,51,52]. HyMovis^®^ displayed a gel-like behavior as well; however, elasticity peculiarly increased with frequency, which was expected as a result of the diverse chemical modification of the biopolymer [12]. The first and third group samples (linear and H-HA/L-HA complexes) behave as typical viscoelastic entangled networks, exhibiting viscous or elastic behavior in certain frequency intervals, depending on the rate of polymeric chains’ disentanglement, which, in turn, is related to biopolymer molecular weight and concentration. Following dilution, as a consequence of the expected increase in relaxation rate, the elastic character of these formulas was observed in a narrower frequency range while, for chemically modified HA samples, it remained predominant throughout the range exploited, entirely covering the physiologically relevant frequencies of knee solicitations [18]. The steady shear pseudoplastic behavior and zero-shear viscosity values for the first and third group gels were consistent with the healthy synovial fluid. Modified HA samples displayed much higher viscosity and resembled only the shear thinning part of the healthy synovial fluid rheological profile [4,5]. Based on the above, chemically modified HA-based preparations are expected to better restore the elastic role of healthy synovial fluid at high-frequency solicitations, but they fail in resembling its viscous behavior at low frequencies. On the other side, the linear and hybrid cooperative complex-based formulas can be predicted to better resemble healthy synovial fluid since they behave as both elastic and viscous material, depending on the frequency of joint solicitation. However, they show reduced performance in the high-frequency region. The maintenance of the rheological profiles after dilution ensures mechanical action even after the naturally occurring mixing of the gels with the pathological synovial fluid upon intra-articular injection, with chemically modified HA samples performing better. The success of the treatment also depends on the maintenance of the rheological action over time. When the rheological performance is retained longer, the therapeutic effect lasts longer and further injections may be delayed. Therefore, more stable products are conceived to be related to higher safety and better patient compliance. Hydrolysis catalyzed by hyaluronidases is one of the main mechanisms responsible for the in vivo degradation of HA, and lately, therefore, in vitro data on resistance to BTH action have been considered useful to predict the relative in vivo longevity of HA-based gels [37,46]. On these bases, we investigated, for the first time, HA-viscosupplement’s ability to retain rheological parameters in the presence of BTH. Except for HyMovis^®^, all products reduced their viscosity in the presence of the enzyme, indicating sensitivity to degradation. On the one side, the higher the degradation, the lower the in vivo rheological performance will probably be. On the other, sensitivity to BTH ensures resorbability, thus avoiding persistence of implants, which may induce inflammation or fibrosis phenomena (e.g., capsule formation, monocyte/cell recruitment). Based on the diverse degradation profiles, Jonexa Hyalastan SGL-80^TM^ is predicted to act longer than the unmodified HA-based formulations. This is rationally related to the cross-linked nature of the biopolymer affecting enzyme recognition/interactions and, thus, degradation kinetics. The specific profile suggests that part of the formula is quickly degraded, leaving a more highly resistant core undergoing slower hydrolysis. This is in line with the coexistence of unmodified and cross-linked HA in the formula as declared by the manufacturer [38]. The first and third group products behave more similarly to each other and to natural HA, showing progressive reductions in viscosity during exposure to BTH. In fact, shorter in vivo permanence is expected for the linear HA-based samples [10,53,54]. It was remarkable that the diverse macromolecule organization in the third group formula resulted in prolonged retention of rheological performance. Under the applied degradative conditions, the Sinovial HL^®^ results were comparable to the cross-linked sample over a wide time range, notwithstanding the absence of chemical modification of natural HA. This peculiar stability, together with the higher HA content, supports the different injection protocol normally suggested for Sinovial HL^®^ in clinical studies (two injections two weeks apart, one from the other, in place of three or five injections, one per week, for traditional protocols). With respect to stability, HyMovis^®^ represents a special case. Its profile suggests the longest therapeutic effect. However, it also indicates that the type and/or extent of HA chemical modifications completely hamper hyaluronidase action (under our experimental conditions), therefore possibly drastically affecting the natural biological pathway of HA and HA-based product degradation. The high resistance to enzymatic hydrolysis is consistent with the chemical modification occurring in the HA carboxyl groups, which are known as important sites for the enzyme activity on HA; this in line with the literature data [37,45,55]. The diverse HA forms were tested in parallel for their bioactivity. As a matter of fact, awareness has been increasing with regard to the crosstalk and biochemical cascade that are potentially activated by HA and products based on its derivatives, in addition to their well-known biomechanical function. Furthermore, the regulatory aspects of these Class III medical devices were recently revised to consider biological aspects. This study, based on primary pathological chondrocytes and synoviocytes made it possible to standardize the biological assays and compare the products on the same reliable experiments. Specifically, we aimed at evaluating the effect of the diverse formulation on cell viability, phenotype preservation, ECM components and inflammatory mediators. Interestingly, all products sustained cell viability in chondrocytes and synoviocytes models, except crosslinked HA (Jonexa Hyalastan SGL-80™), which showed a 40% reduction on synoviocytes. HAS-1 expression was positively modulated, showing the ability to restore joint cells’ functionality, and the hybrid complexes showed the best relative performance. Our results showed an improvement in the specific cartilage biomarkers’ expression at the transcriptional level, such as aggrecan and type II collagen, probably in order to restore the damaged extracellular matrix in OA-compromised chondrocytes. Additionally, our data showed an increase in SOX-9 expression, a specific chondrocyte transcription factor, in the presence of different HA-based gels, among which Sinovial HL^®^ was the most effective. The results obtained on chondrocytes proved that the phenotype was preserved with HA treatments even in the presence of the OA-induced functional deterioration. We recently reported a different pro-inflammatory cytokine modulation due to hybrid cooperative complex treatments [7,15]. Furthermore, Smith and collaborators (2013) [21] investigated the potential mechanism of activity of a modified HA (e.g., HyMovis^®^) compared with an unmodified HA. The authors reported the positive (beneficial) effect of HyMovis^®^ on degrading enzymes (e.g., ADAMTS5 and MMPs), and improvements in terms of inflammation through the reduction in cytokine expression by human joint tissue cells. In the present study, Sinovial HL^®^, HyMovis^®^ gels, and the other products tested, showed a lower MMP-13 mRNA expression, demonstrating efficacy in reducing the degradation of the cartilage matrix. In fact, MMPs, specifically collagenases, are reported to have a key role in rheumatoid arthritis (RA), with OA playing a central role in the matrix fibrillary proteins’ digestion process, specifically collagen [56]. In particular, MMP-13 is involved in matrix destruction through proteoglycans’ degradation (e.g., aggrecan), as described by Mehana and collaborators [57]. However, Pavan and collaborators (2016) [19] reported a lower efficacy in MMPs inhibition for modified HA, (e.g., HYADD), depending on the solubility of the derivatives (e.g., MMP-8 and MMP-13), [19]. Among all HAs tested, hybrid cooperative complexes were the most effective in counteracting inflammation through the NF-kB pathway, as described before, for both chondrocytes and synoviocytes. However, the HA-based preparations investigated in the present study were shown to modulate at the transcriptional level (COMP-2, IL-6, and TNF-α pro-inflammatory proteins) even if, when analyzed by western blotting and IF, the reduction in COMP-2 was superior for Sinovial HL^®^-treated cells than for the other treatments. It is interesting to note that HAS-1 was also modulated by all HA tested, possibly through CD44 [58], but the specific binding ability of modified HA and cross-linked HA has not yet been revealed. It can be argued that the translational feature of this research consists in the possibility of clarifying the potential benefit of each of the HA forms of the diverse groups, highlighting the pros and cons of each product in the in vitro study proposed. For instance, linear HA may be safely used in routine treatments if patient compliance to multiple injections is high, while cross-linked and modified HA may ensure prolonged duration of the viscosupplementation and, thus, a reduced number of injective procedures. However, in this last case, besides a superior elastic behavior that may help to better cope with mechanical stress, there is a concern about the presence of exogenous molecules that, in fact, make HA non-natural, thus reducing the biological effect. Finally, the third group, consisting of hybrid cooperative complexes of high and low MW HA, may better assess at the clinical level traumatic injury or hard inflammation status, also prompting recovery/regeneration. In addition, hybrid cooperative complexes proved stable to a similar extent when compared to cross-linked gels. This may support a clinical protocol of fewer injections and/or prolonged intervals between treatments. In fact, less injective procedures in a clinical approach may be considered beneficial in terms of both patient safety and compliance.

It must be considered that, for a more robust identification of a panel of features to identify diverse groups of formulations, more products should be evaluated for each group. However, even if differences in performance among products belonging to the same group are expected, the study represents a very useful insight on the diverse fingerprint profile of the three analyzed groups, thus representing a useful reference study for both clinicians and scientists involved in the development of novel formulas that are intended for the same purpose.

Another limitation of the study is represented by the accuracy/suitability of prediction of in vivo performance based on relative performance in vitro. A direct clinical comparison of the same products analyzed in vitro would be needed for establishing correct correlation. However, a clinical study comparing diverse products from diverse brands is very unlikely to happen since clinicians generally choose their products according to their experience of reliability. In addition, it is unlikely that a company will support studies which include competitors. For these reasons, independent evaluation with reliable in vitro models is the sole method that currently allows comparative analyses. However, based on recent literature, future research could consider additional in vitro tribological analyses to compare viscosupplements’ frictional properties that, along with rheological features, could more properly describe the overall biophysical action [6,59].

## 5. Conclusions

The potential of the diverse HA forms intended as viscosupplements was evaluated by means of contemporary analysis of their rheological behavior—also after dilution and in the presence of hyaluronidase—and their bioactivity. Compared to the unmodified, linear HA-based product, the chemically modified samples improved mechanical performance during high-frequency solicitation and showed a prolonged viscosupplementation effect at the expense of the biopolymer’s natural chemistry (exogenous molecules are bonded to the biopolymer). The HA hybrid complexes were shown to retain the viscoelastic profile of the first group formula. Further, in the absence of chemical modification, they exhibited comparable stability to the cross-linked sample, thus suggesting prolonged action when delivered into the pathological joints. Modulation of OA-related biomarkers was found for all the HA formulations tested; notably, the third group had a marked anti-inflammatory action accompanied by a major stimulus for hyaluronan synthase on human chondrocytes and synoviocytes. Taken together, these results represent a rather complete comparative panel of in vitro parameters that characterize the diverse HA forms used for intra-articular injections. Data may be a platform for clinicians to better select the most suitable treatment for the specific patient pathology, and for scientists to develop new highly performing HA-formulations addressing specific needs.

## Figures and Tables

**Figure 1 polymers-13-03208-f001:**
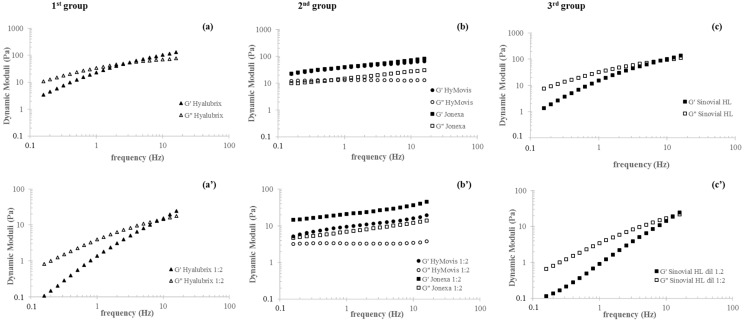
Mechanical spectra for the HA-based formulas as commercialized (**a**–**c**) and after 1:2 dilution in physiological medium (**a’**–**c’**). Measurements were performed at 37 °C.

**Figure 2 polymers-13-03208-f002:**
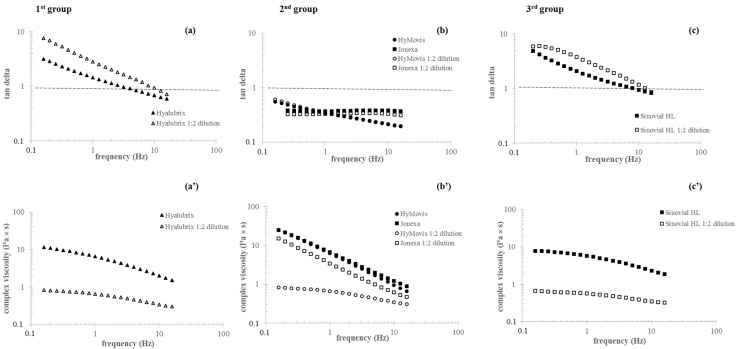
Tan delta and complex viscosity as function of frequency for the HA-based formulas as commercialized (**a**–**c**) and after 1:2 dilution in physiological medium (**a’**–**c’**). Measurements were performed at 37 °C.

**Figure 3 polymers-13-03208-f003:**
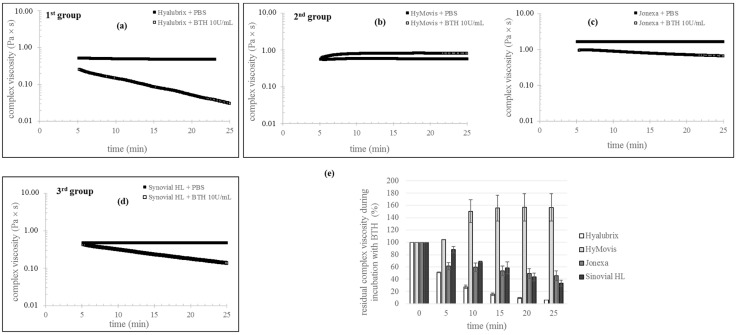
Degradation during incubation with BTH 10 U/mL for the gels (**a**). Reduction in complex viscosity during 25 min incubation with BTH 10 U/mL for the 1st group gel (**a**), the 2nd group gels (**b**,**c**) and the 3rd group formula (**d**). For each gel, the complex viscosity values recorded during incubation with PBS, measured under the same conditions (control), are also reported. Residual complex viscosity (% with respect to the control) at 5, 10, 15, 20, and 25 min of incubation with the enzyme (**e**). The value at time zero (control) is also reported. All measurements were performed at 37 °C, 2% strain and 2.5 Hz frequency.

**Figure 4 polymers-13-03208-f004:**
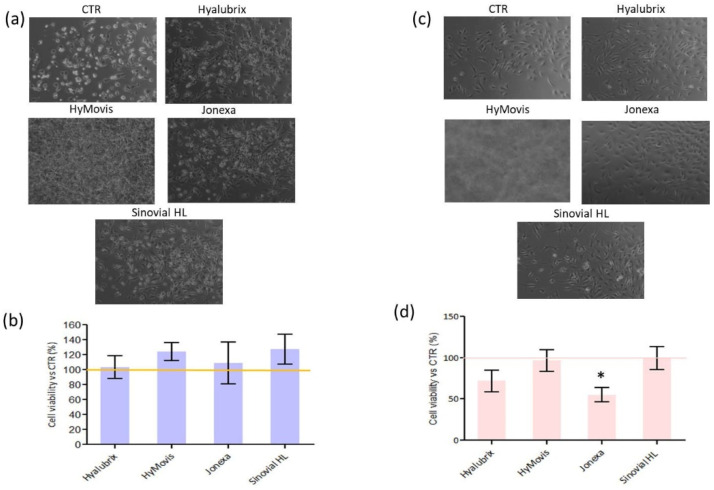
Cell image panels in the presence of different HA hydrogels were compared to untreated chondrocytes (**a**) and synoviocytes (**c**). Quantification of cell viability was performed using CCK8 staining for both chondrocytes (**b**) and synoviocytes (**d**).

**Figure 5 polymers-13-03208-f005:**
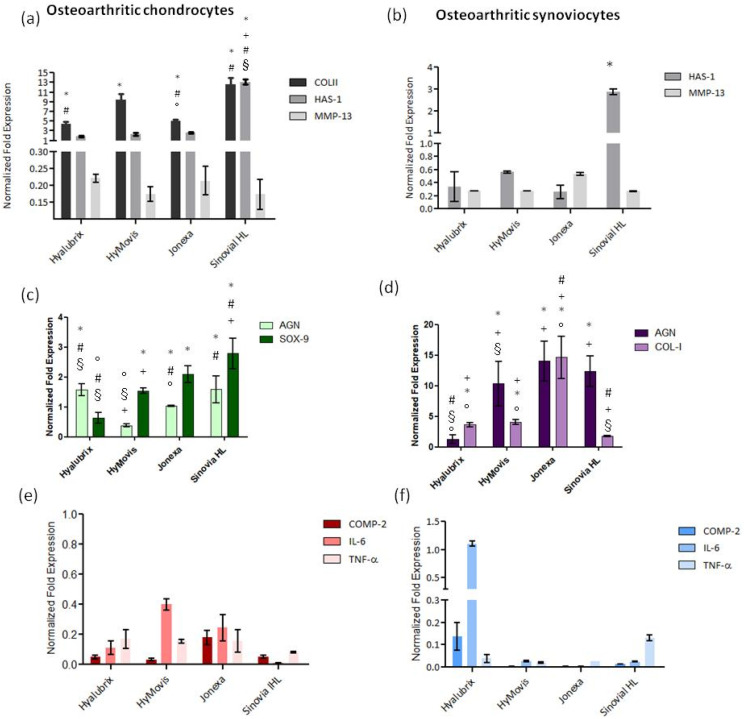
Gene expression analyses normalized with respect to pathological untreated cells (CTR) for COLII, HAS-1, MMP-13 (**a**), AGN and SOX-9 for chondrocytes (**c**), and HAS-1, MMP-13 (**b**), AGN and COL-I for synoviocytes (**d**). In addition, specific inflammation biomarkers (COMP-2, TNF-α, and IL-6) were accomplished for both chondrocytes (**e**) and synoviocytes (**f**). Comparative analyses were performed between a linear HA (Hyalubrix^®^), two chemically modified HAs (HyMovis^®^ and Jonexa Hyalastan SGL-80™), and an HA hybrid cooperative complex (Sinovial HL^®^). The data show the averages to be ± S.D. The statistical significance was analyzed through one-way ANOVA and the Tukey post hoc test for comparison of a family of 5 estimates: * *p* < 0.01 vs. untreated-cells (CTR); # *p* < 0.01 vs. HyMovis^®^, § *p* <0.01 vs. Jonexa Hyalastan SGL-80™; ° *p* < 0.01 vs. Sinovial HL^®^; + *p* < 0.01 vs. Hyalubrix^®^.

**Figure 6 polymers-13-03208-f006:**
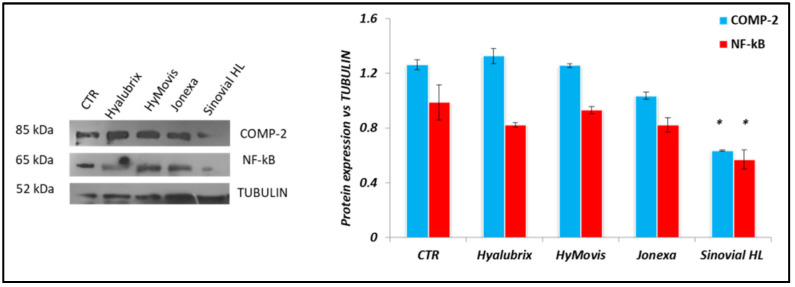
Evaluation of COMP-2 and NF-kB protein levels in pathological chondrocytes treated for 48 h with HA-based gels. Densitometric analysis was performed, normalizing COMP-2 and NF-kB protein expression with respect to TUBULIN: * *p* < 0.05. A *t*-test compared the significance of each treatment with respect to CTR.

**Figure 7 polymers-13-03208-f007:**
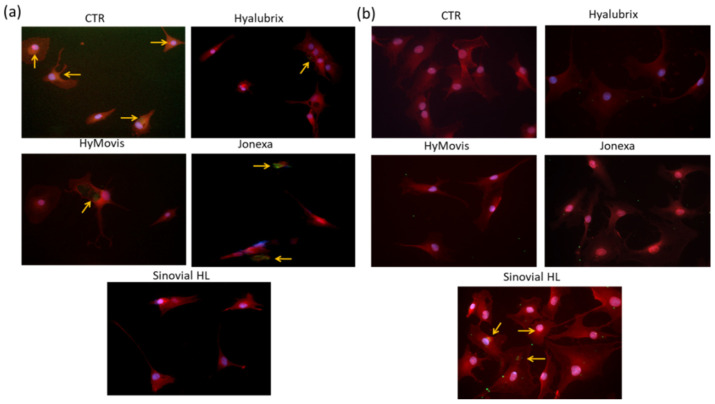
Immunofluorescence staining of COMP-2 (**a**) in treated and untreated primary human chondrocytes; HAS-1 (**b**) in treated and untreated primary human synoviocytes. In blue nuclei, actin fibers of cytoskeleton were stained with tritc phalloidin, and fitc-green antibodies were used for COMP-2 and HAS-1, respectively. Arrows indicate COMP-2 and HAS-1 expression in treated and untreated human primary cells. Pictures were from one representative experiment. Magnification = 40×.

**Table 1 polymers-13-03208-t001:** Available information on the HA-formulations analyzed here, as reported in the leaflets [33,34,38,40,41].

Formulation	HA Amount (Declared on the Label)	Chemical Features	Dose/Syringe Needle Used	Suggested Posology	Classification According to This Research/Paper
Hyalubrix^®^	15 mg/mL	Linear, unmodified HA	18 or 20 G	3 or more injections at weekly intervals	1st group
HyMovis^®^ HYADD 4	8 mg/mL	Modified (derivatized)-HA	18 or 20 G	2 injections at weekly intervals	2nd group
Jonexa Hyalastan SGL-80™	10.5 ± 1 mg/mL	Modified (crosslinked)-HA + unmodified HA	18 or 20 G	1 injection (4 mL) or 2 injections (4 mL + 4 mL) 15 days apart	2nd group
Sinovial HL^®^ 64	16 + 16 mg/mL	Hybrid cooperative complexes NaHyCo	22 G × 1 ½″	2 injections 15 days apart (or differently based on clinical evidence)	3rd group

**Table 2 polymers-13-03208-t002:** Primer sequences used in qRT-PCR.

Gene Name (Symbol)	PCR Primer Sequence 5′→3′	Amplicon Length (bp)
Glyceraldehyde-3-phosphate dehydrogenase (GAPDH)	TGCACCACCAACTGCTTAGC GGCATGGACTGTGGTCATGAG	118
Type II collagen (COLII)	CAACACTGCCAACGTCCAGAT CTGCTTCGTCCAGATAGGCAA	102
Hyaluronan synthase 1 (HAS1)	GGGGATCTTCCCCAAGACC CTCGGAGATTCGGTGGACTA	115
Matrix metallopeptidase 13 (MMP-13)	TCCCTGAAGGGAAGGAGC CTCGTCCAGGATGGCGTAG	105
Cartilage oligomeric protein matrix 2 (COMP-2)	GAGAACTTTGCCGTTGAAGC GCTTCCTGTAGGTGGCAATC	107
Interleukin-6 (IL-6)	GTGGAGATTGTTGCCATCAACG CAGTGGATGCAGGGATGATGTTCTG	112
Tumor necrosis factor alpha (TNF-α)	CGAGTGACAAGCCTGTAGC GGTGTGGGTGAGGAGCACAT	102

**Table 3 polymers-13-03208-t003:** Results of the SEC-TDA characterization. Weight average molar mass (M_w_), numeric average molar mass (M_n_), polydispersity index (M_w_/M_n_), intrinsic viscosity ([η]), and hydrodynamic radius (R_h_) for the water-soluble HA in the tested formulations. The HA concentration, as derived from the analyses, is also indicated. # for hybrid complexes, chromatographic resolution is not complete, which is as expected; thus, the derived parameters can be considered as “apparent” even if they resemble the original HA population used to obtain the complexes through thermal treatment.

Formulation		Hydrodynamic Parameters for the Soluble Fractions					
		M_w_ (kDa)	M_n_ (kDa)	M_w_/M_n_	[η] (dL/g)	R_h_ (nm)	Concentration mg/mL
1st group	Hyalubrix^®^	1180 ± 10	840 ± 10	1.4 ± 0.1	19 ± 1	70 ± 1	14 ± 1
2nd group	HyMovis^®^	n.d.	n.d.	n.d.	n.d.	n.d.	n.d.
	Jonexa Hyalastan SGL-80™	1300 ± 300	800 ± 200	1.6 ± 0.1	20 ± 5	71 ± 1	0.15 ± 0.05
3rd group	Synovial HL 64	1220 ± 20	960 ± 50	1.2 ± 0.1	18 ± 1	70 ± 1	32 ± 1
		140 ± 50	140 ± 10	1.3 ± 0.1	4 ± 1	22 ± 1	

## Data Availability

All data that are critical for reader understanding and discussion of outcomes are reported within the manuscript or as a Appendix A. Additional raw data are available from the corresponding authors upon request.

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
