# Peer review of "Hyaluronan and Derivatives: An In Vitro Multilevel Assessment of Their Potential in Viscosupplementation"

_polymers, 2021, doi:10.3390/polym13193208_

Round 1
Reviewer 1 Report
Lines 12-16 they belong to introduction (justification of the work), but the phrases do not englobe the results of this work. Therefore, it is not abstract.
Line 16: change here for : In this work,....
Line 20-21. The authors mentioned in the abstract the portion of water soluble HA, but the statement was not supported by any reference neither by experimental work.
Line 45-46. A reference is needed.
Line 54. The definition of hyaluronan as "heteropolysaccharide" is quite uncommon. I suggest changing the definition to a more appropiate.
Line 56. The chemical description of HA is wrong. Please correct it
Between lines 59-60 the idea is lost. Please rephrase.
Line 76...Both in Europe and USA the FDA and EMA approved...I think this phrase should be changed to the situation worldwide involving the Asian Market.
The introduction is long and very boring. The hypothesis of this work and the main goal of this work were unfortunately not well described. This might also explain the poor conclusions. It seems that the introduction belongs to a different article because the authors have never explained the reasons of the studies made in this work: Why the enzymes were studied and which were the effect of them. The specificity of the biochemistry of HA.
I think that the authors have to mention clearly why they studied the products. Line 86. it could be important to add information about clinical trials or literature research previous to this work.
A group in Italy (the same country of the authors) mentioned the differences between the marketed products.
https://journals.sagepub.com/doi/10.4137/CMAMD.S38857
Table 1 is missing references. Then, a most deep description of the differences betwen the tested products and the criteria for their study has to be added in the text.
For me, the rheological results were not well discussed. It was necessary to compare with data obtained from clinical trials or research articles of the products. The main question is whether the rheological results are predicting the clinical outcomes...but I dont think so.
The same problematic was discussed in the next article.
https://journals.plos.org/plosone/article?id=10.1371/journal.pone.0216702
Also, the in vitro data are representative for the in vivo situation.
I think that it was better to use standarized cell cultivations.
Example see the next article
https://pubmed.ncbi.nlm.nih.gov/33466397/
About the degradation of hyalurodinase, it is important to mention the pharmacopeia conditions.
Line 398 a reference is missing.
Line 559 a reference is missing.
Line 581-585. they dont belong in here. Probably, this part should be moved to the introduction.
Many parts of this manuscript are very vague. They were based on speculations or in the references from the same authors. I suggest adding more relevant literature.
For example, the effect of hymovis was studied by Smith et al. This reference was not included neither discussed to validate the results of the present study.
https://journal-inflammation.biomedcentral.com/articles/10.1186/1476-9255-10-26
For example, the inhibition of MMP13 was already described. This reference has to be added on line 400.
In general, the authors used home-made protocols for in vitro test which is hard to know whether they can be validated. I am also afraid of the reproducibility of the data because the process for the cells extraction was not well defined.
It could be good to add a scheme/table explaining the mechanism of action of HA, because the text is very diffucult to analyze and follow.
It is very hard to validate the preliminary results that the authors reported. Moreover, when the efficacy of HA on viscosupplementation has not been demonstrated in-vitro or in vivo. The only available results on the literature are based on injections of HA in clinical trials on probands compared to placebo. Therefore, I think that this article is not explaining anything new in the field. Moreover, when the authors tested very different products, without a clear explanation of why they studied them.
Author Response
Thank you for your suggestions we believe your critical review has been very helpful to improve the manuscript.
- Lines 12-16 they belong to introduction (justification of the work), but the phrases do not englobe the results of this work. Therefore, it is not abstract.
Ans.:The abstract has been improved. Thank you for the suggestion.
2.Line 16: change here for : In this work,....
Ans.: We replaced “here” with “in this work”.
- Line 20-21. The authors mentioned in the abstract the portion of water soluble HA, but the statement was not supported by any reference neither by experimental work.
Ans.: The quantitative determination of the water-soluble fraction of HA included in the samples along with its hydrodynamic analysis was reported in the text (paragraph 2.2.1). The paragraph was modified to better explain the protocol.
- Line 45-46. A reference is needed.
- Line 54. The definition of hyaluronan as "heteropolysaccharide" is quite uncommon. I suggest changing the definition to a more appropiate.
6.Line 56. The chemical description of HA is wrong. Please correct it
7.Between lines 59-60 the idea is lost. Please rephrase.
Ans.: All these points were considered in the revised version of the Introduction section
8.Line 76...Both in Europe and USA the FDA and EMA approved...I think this phrase should be changed to the situation worldwide involving the Asian Market.
The referee is correctly indicating that there is a huge market in Asia. We modified the introduction as follows:
“The increasing number of commercialized products (EMA or FDA approved and/or registered in Asia and Africa), does not permit a full array comparison between all of them “
9.The introduction is long and very boring. The hypothesis of this work and the main goal of this work were unfortunately not well described. This might also explain the poor conclusions. It seems that the introduction belongs to a different article because the authors have never explained the reasons of the studies made in this work: Why the enzymes were studied and which were the effect of them. The specificity of the biochemistry of HA.
I think that the authors have to mention clearly why they studied the products. Line 86. it could be important to add information about clinical trials or literature research previous to this work.
A group in Italy (the same country of the authors) mentioned the differences between the marketed products.
https://journals.sagepub.com/doi/10.4137/CMAMD.S38857
Ans: Comments 4-9 refer to the introduction. The section has been extensively rearranged and shortened to fulfil the referee requirements
We thank the referee for providing us with this link above. The manuscript was considered in the revision of the Introduction. The above mentioned report demonstrated, through a systematic analysis of literature reports on HA-based products (commercialized in Italy) intended for intrarticular injections, that the scientific evidence on these products is scarce. The authors finally highlighted that there is a strong need for larger and brand-specific studies to increase awareness and correct use of these devices. The in vitro characterization we carried out on some specific products, selected among the unmodified HA-based, chemically modified HA-based and complexed HA-based products, represents a step in this direction. We inserted this reference in the introduction section to better present the work.
- Table 1 is missing references. Then, a most deep description of the differences between the tested products and the criteria for their study has to be added in the text.
Ans.: Information reported in Table 1 were derived from the leaflets of the products. This was reported in the caption of the table. The references have been added in the revised version of the manuscript. The reason/criteria for the study of these products were better explained in the revised introduction.
- For me, the rheological results were not well discussed. It was necessary to compare with data obtained from clinical trials or research articles of the products. The main question is whether the rheological results are predicting the clinical outcomes...but I dont think so.
The same problematic was discussed in the next article.
https://journals.plos.org/plosone/article?id=10.1371/journal.pone.0216702
Ans.: We agree with the referee that discussing the relative rheological data in relation to data from clinical trials on the tested products would be very useful. However, in vivo data from clinical trials comparing all the products tested here, within the same experimental set up, would be needed. In fact, a reliable and correct comparison, would be ensured only in this specific case.
The main limitation for establishing the relation between relative rheological behavior of viscosupplements and their in vivo effects is the lack of reliable clinical comparison. More frequently, because there are not scientific reports of extensive comparisons between medical devices in clinical studies (only as meta-analyses), in vitro data on different products are compared considering aggregated in vivo data from different studies with diverse experimental set up. This may be misleading and may be the reason for the controversy occurring in literature about 1. clinical efficacy of the viscosupplementation treatments and 2. relation between clinical outcomes and products rheological properties (Henrotin, Semin Arthritis Rheum. 2015,45(2):140-9; Legré-Boyerabc, Orthopaedics & Traumatology: Surgery & Research, 2015, S101-S108; Bonnevie, PLoSONE 14(5):e0216702; ).
Since no clinical trials comparing the products tested in this work are currently available, we limited our discussion to the comparison of the rheological profiles derived for the tested formulations (in the same experimental set up), as such and after dilution, to the typical profile of healthy synovial fluid. Viscosupplementation is intended to restore the synovial fluid rheological behavior, that is reduced under pathological conditions therefore, this comparison allows us to predict potential differences among the preparations in restoring synovial fluid viscoelasticity. This type of comparison is already reported in the literature and discussion of the potential viscosupplementation action of a preparation, based on its in vitro rheological data, is widely accepted (Borzacchiello, J Biomed Mater res 2010, 92A, 1162; Mathieu Clin Orthop Realt Res 2009, 46, 3002; Bhuanantanondh J Med Biol Eng 2010, 32(1), 12; Finelli, Biorheology 2011, 48, 263; Lapasin Chem. Biochem. Eng. Q. 2015, 29(4), 511;). For instance, it is acknowledged that the rheological behavior measured at 0.5 and 2.5Hz frequency may represent a way to evaluate how synovial fluid and viscosupplements react to walking and running conditions.
The referee’s comment highlighted another important issue, related to the in vitro features of viscosupplements that better predict in vivo performance. The controversy existing on the correlation between in vitro rheological properties and in vivo performance was discussed above. We are aware that very recently compared to the rheological investigation, tribological analyses have been also performed on viscosupplements to study their effect on the friction of the articular cartilage (Bonnevie, PLoSONE 14(5):e0216702; Rebenda Tribology Int 2021, 160, 107030). As indicated by the referee, such studies have been proposed by Bonnevie et al. (2019) as a better mean to predict formulations in vivo behavior. However, the cited study suffers from the same limitation above mentioned since clinical data are collected from diverse independent studies not contemporary involving the products compared in vitro.
Rationally, we are still far to ascertain whether rheological or tribological characteristics of the preparations are the best for predicting in vivo outcomes also considering there are still many ambiguities about the action mechanism. However, both methodologies are currently recognized as useful for the purpose with the rheological analyses representing the most used up to date.
We added some sentences in the text to discuss the overall limitations of the study, taking into account the diverse issues emerging from this comment (page 17 e lines 634-651).
- Also, the in vitro data are representative for the in vivo situation.
I think that it was better to use standarized cell cultivations.
Example see the next article
https://pubmed.ncbi.nlm.nih.gov/33466397/
Ans.: Thanks for this consideration.
We are aware of the simplification of the in vitro model based on 2D cultures. We are aware of the complexity of OA pathophysiological mechanism, as it can be found in the recent scientific literature different in vitro and in vivo models have been developed over the years. However, isolating and culturing primary cells represent a well-established, cost-effective method to obtain reliable and high throughput result. Based on the literature, and considering a quite long experience in cell cultures, we attempted few years ago in establishing a robust in vitro model for OA. The strategy included a direct collaboration between biochemists and orthopaedical surgeons. The stage of OA in the patient, the kind of tissue/part of the cartilage/joint, removed during the surgical approach, the solution used to preserve it, the time interval before tissue digestion and cell separations, buffers, enzyme amount, and even washing from blood cells were accurately standardized. A Specific standard operating procedure (SOP) was described, and these protocols are run from the same persons since 2015, Cell Sorter and biomarkers evaluation are generally used in parallel to morphologycal analyses. (some of the results obtained with these models have been already published and the references are present both in materials and methods section and in the discussion- (e.g. Stellavato A, et al. J Cell Biochem. 2016 Sep;117(9):2158-69. doi: 10.1002/jcb.25556. Epub 2016 May 11. PMID: 27018169; PMCID: PMC5084766, and Stellavato A, et al. Biomed Res Int. 2019 Apr 23;2019:4328219. doi: 10.1155/2019/4328219. Erratum in: Biomed Res Int. 2020 Jul 25;2020:7530149. PMID: 31179322; PMCID: PMC6507116, Vassallo V. Unsulfated biotechnological chondroitin by itself as well as in combination with high molecular weight hyaluronan improves the inflammation profile in osteoarthritis in vitro model. J Cell Biochem. 2021 May 31. doi: 10.1002/jcb.29907. Epub ahead of print. PMID: 34056757).
We agree with the referee, each patient has a different biological fingerprint, however the stage of OA is consistent between donors, and however, the variability of patients is consistent with the one found in clinical practice. The authors preferred to establish this human primary cells model rather than referring to animal tissues.
- About the degradation of hyalurodinase, it is important to mention the pharmacopeia conditions.
Ans.: Whilst Pharmacopeia methods are the gold standard for GMP products release, the possibility to explore other assays and compare an array of different test condition is widening the knowledge in the field. In fact, very often pharmacopeia tests are referred to techniques that are estabilished since many years notwithstanding new technical possibilities due to novel equipment release. Thus from a research viewpoint, pharmacopeia tests are basilar ones to be improved and challenged to increase knowledge about products. In fact, very often novel validated protocols are proposed not only by research institutions but also by R&D departments of companies
Despite there is a long experience in BTH studies, more recently exploiting the SEC—TDA analyses to provide an extensive hydrodynamic analyses of fragments (BMC Cell Biology (2015) 16:19
DOI 10.1186/s12860-015-0064-6; Anal Biochem. 2010 Sep 1;404(1):21-9; International Journal of Biological Macromolecules 84 (2016) 221–226) we here preferred to obtain a comparative analyses on rheological performance since it seemed to better comply for the intended purposes of the formulations.
- Line 398 a reference is missing.
Line 559 a reference is missing.
References were inserted as requested
The results obtained here are in agreement with previous ones. Specifically hybrid complexes were compared to linear HA and/or lightly crosslinked HA in Merola F, Scrima M, Melito C, Iorio A, Pisano C, Giori AM, Ferravante A. A novel animal model for residence time evaluation of injectable hyaluronic acid-based fillers using high-frequency ultrasound-based approach. Clin Cosmet Investig Dermatol. 2018 Jul 11;11:339-346. doi: 10.2147/CCID.S156740. PMID: 30022845; PMCID: PMC6045909.
- Line 581-585. they don’t belong in here. Probably, this part should be moved to the introduction.
Ans: Thank you for the suggestion. The part has been cancelled from the discussion secion.
- Many parts of this manuscript are very vague. They were based on speculations or in the references from the same authors. I suggest adding more relevant literature.
For example, the effect of HyMovis was studied by Smith et al. This reference was not included neither discussed to validate the results of the present study.
https://journal-inflammation.biomedcentral.com/articles/10.1186/1476-9255-10-26
For example, the inhibition of MMP13 was already described. This reference has to be added on line 400.
Ans.:The references were inserted and discussed as requested, we thank the referee for suggesting these literature reports.
- In general, the authors used home-made protocols for in vitro test which is hard to know whether they can be validated. I am also afraid of the reproducibility of the data because the process for the cells extraction was not well defined.
The protocols were already described and used in the last 6 years in our lab. They are at the basis of publications in international journals with quite high IF (e.g. Stellavato A, et al. J Cell Biochem. 2016 Sep;117(9):2158-69. doi: 10.1002/jcb.25556. Epub 2016 May 11. PMID: 27018169; PMCID: PMC5084766, and Stellavato A, et al. Biomed Res Int. 2019 Apr 23;2019:4328219. doi: 10.1155/2019/4328219. Erratum in: Biomed Res Int. 2020 Jul 25;2020:7530149. PMID: 31179322; PMCID: PMC6507116, Vassallo V. Unsulfated biotechnological chondroitin by itself as well as in combination with high molecular weight hyaluronan improves the inflammation profile in osteoarthritis in vitro model. J Cell Biochem. 2021 May 31. doi: 10.1002/jcb.29907. Epub ahead of print. PMID: 34056757, Stellavato A, Restaino OF, Vassallo V, Cassese E, Finamore R, Ruosi C, Schiraldi C. Chondroitin Sulfate in USA Dietary Supplements in Comparison to Pharma Grade Products: Analytical Fingerprint and Potential Anti-Inflammatory Effect on Human Osteoartritic Chondrocytes and Synoviocytes. Pharmaceutics. 2021 May 17;13(5):737. doi: 10.3390/pharmaceutics13050737. PMID: 34067775; PMCID: PMC8156081. Stellavato A, Restaino OF, Vassallo V, Finamore R, Ruosi C, Cassese E, De Rosa M, Schiraldi C. Comparative Analyses of Pharmaceuticals or Food Supplements Containing Chondroitin Sulfate: Are Their Bioactivities Equivalent? Adv Ther. 2019 Nov;36(11):3221-3237. doi: 10.1007/s12325-019-01064-8. Epub 2019 Sep 7. PMID: 31494830; PMCID: PMC6822805, Russo R, Vassallo V, Stellavato A, Valletta M, Cimini D, Pedone PV, Schiraldi C, Chambery A. Differential Secretome Profiling of Human Osteoarthritic Synoviocytes Treated with Biotechnological Unsulfated and Marine Sulfated Chondroitins. Int J Mol Sci. 2020 May 26;21(11):3746. doi: 10.3390/ijms21113746. PMID: 32466468; PMCID: PMC7312545, Stellavato A, Pirozzi AVA, de Novellis F, Scognamiglio I, Vassallo V, Giori AM, De Rosa M, Schiraldi C. In vitro assessment of nutraceutical compounds and novel nutraceutical formulations in a liver-steatosis-based model. Lipids Health Dis. 2018 Feb 5;17(1):24. doi: 10.1186/s12944-018-0663-2. PMID: 29402273; PMCID: PMC5800044).
Our lab is certified ISO9001, and inspected every year since 2009. We have reliable Standard operating procedures also for the specific difficult task to work with primary cells. However, we preferred to move to these in vitro models. In fact, especially for joint derived cells, cell lines often appeared to be de-differentiated thus molecular outcomes are, in our opinion, less or even not nreliable.
Referees may find these kind of consideration in our previous work.
We have worked with over 100 samples from the surgical procedures. Especially in the last 5 years we have established a specific specimen selection with the orthopaedic surgeons, and we use whenever possible (sufficient number of cells) cell sorter characterization or, on the basis of previously established correlation, col-II expression for chondrocytes as specific markers is quantified by RT-PCR with respect to ColI-I)
Despite several disadvantages of in vitro cell culture, such as high variability (sometimes different passages), de-differentiation and changes in morphology, numerous are the advantages. Using primary cells permittedis to work on a well-established, cost-effective method to obtain reliable and high throughput result. Also, cell cultures permit a fine control of the physicochemical environmental conditions (i.e., pH, temperature, osmotic pressure, oxygen, and carbon dioxide tension). Also, in consideration of the 3R approach (refinement, reduction and replacement), the in vitro model based on primary cells seems the best alternative.
For our in vitro experiments we performed a phenotypic Characterization of Articular Chondrocytes and synoviocytes, through Fluorescence-Activated Cell Sorting (FACS), as reported by Stellavato A, et al. J Cell Biochem. 2016 Sep;117(9):2158-69. doi: 10.1002/jcb.25556. Epub 2016 May 11. PMID: 27018169; PMCID: PMC5084766, and Stellavato A, et al. Biomed Res Int. 2019 Apr 23;2019:4328219. doi: 10.1155/2019/4328219. Erratum in: Biomed Res Int. 2020 Jul 25;2020:7530149. PMID: 31179322; PMCID: PMC6507116.
- It could be good to add a scheme/table explaining the mechanism of action of HA, because the text is very diffucult to analyze and follow.
Ans.: Whilst the effect of the viscosupplements in restoring the biomechanical features of healthy synovial fluid is well established and the biophysical characterization here reported can be analysed in view of the previous literature reports.
As far as concern the biochemical and biological activity, that has been more recently approached for HA intrarticular injection, we aimed at evaluating a general effect on synoviocytes that are the major cell population in the joint (sinovial fluid + synovial membrane), and also on chondrocytes that are in fact damaged in the OA pathology, and exposed to the inflamed synovial. Considering that the eventually occurring repair mechanism should include: reduction of the inflammation, cell viability (duplication) and phenotype maintenance and biosynthesis of the key macromolecules for extracellular matrix features, we attempted in analysing and quantifying all these mediators.
The description of the specific biomarkers role in relation to the pathology (or in the action to counteract detrimental condition of OA) have been described in the discussion section (lines 581-603).
- It is very hard to validate the preliminary results that the authors reported. Moreover, when the efficacy of HA on viscosupplementation has not been demonstrated in-vitro or in vivo. The only available results on the literature are based on injections of HA in clinical trials on probands compared to placebo. Therefore, I think that this article is not explaining anything new in the field. Moreover, when the authors tested very different products, without a clear explanation of why they studied them.
Ans.: We think that this comment was addressed by the revised introduction, and the expandend discussion
The viscosupplementation effect is recognized by the guidelines of well established international organization as reported by:
Conrozier T, Monfort J, Chevalier X, Raman R, Richette P, Diraçoglù D, Bard H, Baron D, Jerosch J, Migliore A, Henrotin Y. EUROVISCO Recommendations for Optimizing the Clinical Results of Viscosupplementation in Osteoarthritis. Cartilage. 2020 Jan;11(1):47-59. doi: 10.1177/1947603518783455. Epub 2018 Jun 21. PMID: 29926748; PMCID: PMC6921960.
(Henrotin Y, Raman R, Richette P, Bard H, Jerosch J, Conrozier T, Chevalier X, Migliore A. Consensus statement on viscosupplementation with hyaluronic acid for the management of osteoarthritis. Semin Arthritis Rheum. 2015 Oct;45(2):140-9. doi: 10.1016/j.semarthrit.2015.04.011. Epub 2015 Apr 30. PMID: 26094903.
Therefore our manuscript was not intended to clarify if HA gels are effective in treatments. Considering the large amount of injective intrarticular procedures, that are increasing over time, also due to aging and obesity, we were considering to characterize products (and also diverse technological approache e.g chemical modification) on the biophysical and biochemical features to allow reliable comparison with the same methods. In fact, these kind of comparative studies are lacking, specifically considering the 3 forms of HA here arranged in categories.
As a matter of fact our data may contribute in increasing knowledge about products, and this should be helpful for clinicians, and maybe supportive for patients consciousness about treatments.
The referee may disagree, but I think we better explained now our approach.

Reviewer 2 Report
" Hyaluronan and Derivatives: an in vitro Multilevel Assessment of Their Potential in Visco supplementation"
It is interesting to investigate the comparison, using extensive rheological analyses, hydrodynamic parameters and susceptibility to enzymatic degradation, representative products from these three different groups. This manuscript is very well written. However, there are a few corrections that are essential to meet the standard for publication. Please refer to the following comments.
- Please add about the statistical evaluation method in this study. If you are using the software, please add the name and version of the statistical software. If you used the analysis by the original code (Python etc.), please add the library and version you use.
- The authors compare the physical characteristics of hyaluronic acid and its derivatives using various methods. Using a multifaceted analysis method is a very good method. However, we need to mention the limitations of these research methods and results. Please add it to the discussion section.
Author Response
" Hyaluronan and Derivatives: an in vitro Multilevel Assessment of Their Potential in Visco supplementation"
It is interesting to investigate the comparison, using extensive rheological analyses, hydrodynamic parameters and susceptibility to enzymatic degradation, representative products from these three different groups. This manuscript is very well written. However, there are a few corrections that are essential to meet the standard for publication. Please refer to the following comments.
- Please add about the statistical evaluation method in this study. If you are using the software, please add the name and version of the statistical software. If you used the analysis by the original code (Python etc.), please add the library and version you use.
Differences in real time PCR results are shown as the averages ± S.D. The statistical significance was analyzed through one-way ANOVA and the Tukey post hoc test for comparing a family of 5 estimates as reported in the manuscript. For western blotting experiments t-student test was performed.
For rheology measurements, as reported, analyses were carried out in duplicate and presented as average of the two. Therefore, statistical analyses were not reported. However, results from the two measurements were consistent, with differences in values less than 10% over the wide range of clinically relevant frequencies. The authors compared their measurements with the ones reported in literature, showing consistency (e.g. Hyalubrix, Russo et al., PLoS ONE 11(6): e0157048)
- The authors compare the physical characteristics of hyaluronic acid and its derivatives using various methods. Using a multifaceted analysis method is a very good method. However, we need to mention the limitations of these research methods and results. Please add it to the discussion section.
Ans: It has to be considered that for a more robust identification of a panel of features identifying the diverse groups of formulations, more products for each group should be considered. However, even if (quantitative) differences in behavior among products belonging to the same group are expected, the study represents a very useful insight to obtain a fingerprint of the three analyzed groups thus representing a useful reference study for both clinicians, and scientists involved in the development of novel formulas intended for the same purpose.
Another limitation of the study is represented by the accuracy/suitability of prediction of in vivo performance of the formulas based on their relative in vitro features. A direct clinical comparison of the same products analyzed in vitro would be needed to establish a robust correlation. However, a clinical study comparing diverse products from diverse brands is very unlikely to happen since clinicians are generally choosing their products according to their experience of reliability. Furthermore, it is unlikely that a company will support studies which include competitors. For these reasons, independent evaluation with in vitro reliable models are the sole that currently allow a valid comparative analyses. However, based on recent literature, future research could consider additional in vitro tribological analyses to compare viscosupplements’ frictional properties that, along with rheological features, could more properly describe the overall biophysical action (Bonnevie PLoSONE 14(5): e0216702; Rebenda et al. Tribology International 2021, 160,107030).
Thank you for this comment: we inserted some sentences in the text to discuss limitations of the work (page 17, lines 634-651).

Reviewer 3 Report
Dear authors,
Here are my comments for this paper. The manuscript is clearly written and data are consistent.
- Please clearly show the novelty and addev value of the manuscript with respect to existent literature. You mentnioned something in the introduction, but it could be strenghten.
- Why did you choose 1.59 Hz as frequency for the amplitutde sweep test?
- Why did you choose 0.159–15.9 Hz domanin for oscillation tests?
- I think the number of references could be improved with more relevant papers.
Author Response
Dear authors,
Here are my comments for this paper. The manuscript is clearly written and data are consistent.
- Please clearly show the novelty and addev value of the manuscript with respect to existent literature. You mentnioned something in the introduction, but it could be strenghten.
Ans.: The introduction was extensively revised to better present the aim of the work and the added value in respect to the current literature. Thank you for the comment.
- Why did you choose 1.59 Hz as frequency for the amplitutde sweep test?
- Why did you choose 0.159–15.9 Hz domanin for oscillation tests?
Ans(Q2;Q3): 10 rad/s angular frequency, corresponding to 1.59Hz frequency, is a typical frequency value used for amplitude sweep tests. We inserted some specific references in the manuscript (Russo et al., PLoSONE 11(6): e0157048; Zhao X, Wang L, Gao J, Chenc X, Wang K. Hyaluronic acid/lysozyme self-assembled coacervate to promote cutaneous wound healing. Biomater Sci 2020; 8: 1702–1710. DOI: 10.1039/c9bm01886g; Cao J, Kang Y, Xiaoqing, Wu, He C, Zhou J. Self-healing and easy to shape mineralized hydrogels for iontronics. J Mater Chem B 2020; 1-7. DOI: 10.1039/x0xx00000x; Wu G, Jinab K, Liu L, Zhang H. Rapid self-healing hydrogel based on PVA and sodium alginate with conductive and cold-resistant property. RSC Adv 2020; 1-3. DOI: 10.1039/C9SM02455G).
For the same reason we studied the moduli dependence over a frequency range varying from 1 rad/s to 100rad/s (Bonnevie et al. PloSONE 14(5):e0216702; Russo et al., PLoSONE 11(6): e0157048). This range encompasses the physiological frequencies of knee movements (walking and running, respectively) thus resulting appropriate for the specific study. Based on the comment of the referee, we modified the unit in rad/s (see Materials and Methods, page 4 lines 141 and 145 ) that may cause confusion. Thank you for the observation.
- I think the number of references could be improved with more relevant papers.
Ans.: References have been added both in the introduction and discussion section to better present and discuss the work.

Round 2
Reviewer 1 Report
The authors made a great work and included several references in order to discuss/explain the misunderstandings in the first version of the manuscript. I do agree with the publication of the work.
The topic is quite complex, but it was very well discussed by the authors. I do recommend acceptance of this work.
Reviewer 2 Report
Thank you for giving me this opportunity to re-review your revised manuscript.
I am happy that all of the suggested corrections have been made.
Thank you for spending so much time for revised manuscript.